# EYERUBBICS: The Eye Rubbing Cycle Study

**DOI:** 10.3390/jcm12041529

**Published:** 2023-02-15

**Authors:** Alexandre Hage, Juliette Knoeri, Loïc Leveziel, Alexandre Majoulet, Jean-Victor Blanc, Juliette Buffault, Antoine Labbé, Christophe Baudouin

**Affiliations:** 1Department of Ophthalmology III, National Hospital Centre for Ophthalmology Quinze-Vingts, IHU FOReSIGHT, 28 Rue de Charenton, F-75012 Paris, France; 2Department of Ophthalmology V, National Hospital Centre for Ophthalmology Quinze-Vingts, IHU FOReSIGHT, 28 Rue de Charenton, F-75012 Paris, France; 3Department of Adult Psychiatry and Medical Psychology, Saint-Antoine Hospital, Sorbonne Université, AP-HP, 184 Rue du Faubourg Saint-Antoine, F-75012 Paris, France; 4Department of Ophthalmology, Ambroise Paré Hospital, AP-HP, Université de Versailles Saint-Quentin en Yvellines, 9 Avenue Charles de Gaulle, F-92100 Boulogne-Billancourt, France; 5INSERM, CNRS, Institut de la Vision, Sorbonne Université, 17 Rue Moreau, F-75012 Paris, France

**Keywords:** addiction, dry eye disease, eye rubbing, itch, keratoconus

## Abstract

**Purpose**: Describe the cognitive and behavioral symptomatology of patients with chronic ocular rubbing in keratoconus (KC) and Ocular Surface Disease (OSD) using a self-questionnaire. **Methods**: A prospective study was conducted in a tertiary ophthalmology center between May and July 2021. We consecutively included all patients presenting with one of the following conditions: KC and OSD. A questionnaire including the evaluation of Goodman and CAGE-modified criteria for eye rubbing was given to patients in consultation to evaluate their ocular symptoms and medical background. **Results**: We included 153 patients in the study. Of these, 125 (81.7%) patients reported eye rubbing. The average Goodman score was 5.8 ± 3.1 and was ≥ 5 in 63.2% of cases. The CAGE score was ≥ 2 in 74.4% of patients. Addiction (*p* = 0.045) and psychiatric family history (*p* = 0.03) were more frequent in patients with higher scores. Ocular symptoms and eye rubbing were significantly more frequent and intense in patients with higher scores; **Conclusion**: Eye rubbing presents addictive-like cognitive and behavioral characteristics in patients with KC or OSD. The eye rubbing cycle could play an essential role in the onset and progression of keratoconus and could be a factor in the maintenance of dry eye.

## 1. Introduction

Pruritus is clinically defined as the subjective and unpleasant sensation of a desire to scratch [1]. Many ophthalmologic conditions cause ocular pruritus, including allergic conjunctivitis, dry eye syndrome and blepharitis [2]. A study of 400 patients consulting an ophthalmology center in Philadelphia found a prevalence of chronic pruritus of 29.5%, with a majority of allergic conjunctivitis (51.8%), dry eye syndrome (40.1%) and blepharitis (43.8%) [3]. This study reported a higher prevalence of pruritus in women. Moreover, the intensity of ocular pruritus was correlated with the duration of its evolution, suggesting a phenomenon of neuronal sensitization as in other types of pruritus [4].

Eye rubbing is a frequent behavioral manifestation. It occurs in most cases in response to the sensation of ocular pruritus or irritation to relieve an unpleasant sensory stimulus. It can also occur spontaneously, particularly at bedtime or upon awakening, during ocular fatigue or even be psychogenic [5]. The prevalence of eye rubbing in keratoconus varies between 66 and 91%, and it has been suggested that it could in fact be a causal factor of the disease [6]. As for other Ocular Surface Diseases (OSD) such as dry eye syndrome, a study estimates the prevalence of ocular rubbing at around 20 to 30% [7]. Ocular rubbing constitutes a physical stress exerted on the eye and its appendages. This stress induces numerous modifications: variations of the IOP, biomechanical, cellular and biochemical consequences, occurrence of corneal ectasia, etc. [8,9,10].

The pathophysiology of pruritus and ocular rubbing is already a subject of research [11]. However, no study has addressed the cognitive–behavioral aspect of this symptomatic duo. In dermatology, chronic pruritus is considered to be part of an “itch–scratch” cycle [12]. Cognitive–behavioral mechanisms parallel to those found in addictions have been identified for scratching [13,14]. As for cutaneous pruritus, we hypothesized that ocular pruritus and eye rubbing are part of a vicious cycle with cognitive–behavioral bases common to addictive disorders.

The aim of this non-interventional study was to describe, using a self-questionnaire, the cognitive and behavioral symptomatology of patients with chronic eye rubbing in a population of keratoconus and OSD.

## 2. Materials and Methods

We conducted this single-center prospective study in a tertiary ophthalmology center (Centre Hospitalier National d’Ophtalmologie des Quinze-Vingts, Paris, France) between May 2021 and July 2021. We included over two consecutive months all patients presenting to ophthalmology consultations in two specialized consultations and presenting with one of the following pathologies: keratoconus (KC group) and Ocular Surface Disease (OSD group). The exclusion criteria were as follows: age younger than 18 years, history of recent eye surgery (<3 months) and presence of a language barrier preventing reliable responses. Patients who incorrectly completed the questionnaire (e.g., missing page) were excluded retrospectively. The questionnaire was given to patients in consultation on a voluntary basis. Informed consent was obtained from the subjects after explanation of the nature of the study. The research followed the tenets of the Declaration of Helsinki.

The first part of the questionnaire included an interrogation concerning the patient’s ophthalmological, dermatological and psychiatric background and a self-evaluation of ophthalmological symptoms. The following data were collected: age, sex, working conditions, medical and surgical ophthalmological history, ophthalmological treatments, use of glasses and contact lenses. Next, we assessed the ocular symptoms experienced by the patients using an open-ended question. Symptomatology was then evaluated in terms of frequency and intensity using the SANDE (Symptom Assessment In Dry Eye) questionnaire, which estimates the frequency of occurrence of symptoms and their intensity [15]. Psychiatric history was sought: addictive disorder, personal psychiatric disorder or family history. We looked for the presence of dermatological history as well as the existence of skin pruritus and scratching.

The second part of the questionnaire dealt with eye rubbing. As for the ocular symptoms, we used the SANDE scale to evaluate the frequency and intensity of eye rubbing. We looked for daily or occasional occurrences as well as the period of day. We used the Goodman and DSM-5 diagnostic criteria for addiction to obtain a 16-item cognitive–behavioral assessment of eye rubbing [16]. We chose the limit of 5 positive responses as an indicator of addictive-like behavior. The CAGE questionnaire used to predict the existence of an alcohol use disorder was also submitted to the patient with four questions adapted to eye rubbing [17]. We considered the presence of 2 positive responses to the CAGE questionnaire as predictive of an addictive-like disorder related to eye rubbing. Finally, we asked patients if they intended to stop eye rubbing after completing the questionnaire. The primary endpoint was the assessment of Goodman criteria for patients with eye rubbing [16]. Secondary endpoints were the evaluation of CAGE score results, psychiatric and addictive comorbidities.

An ophthalmologic examination by a specialist was performed on the same day for each patient. The following data were collected for all patients: best corrected distance visual acuity (CDVA), intraocular pressure (IOP), presence of follicles, papillae or blepharitis, break-up time (BUT) and Oxford grading score (Oxf). For keratoconus, we evaluated the presence of Vogt’s striae and corneal opacities, the maximum keratometry (Kmax) and the thinnest pachymetry (measured on Pentacam, Oculus).

Statistical analysis of the data was performed with MedCalc^®^ software v20.218 (MedCalc Software Ltd.^®^, Ostend, Belgium). Chi-square test, Student *t*-test, ANOVA test, and logistic regression were used. Variables were first studied in univariate analysis and then analyzed in multivariate regression. Regression analysis was performed using Spearman’s correlation coefficient. The threshold for statistical significance was set at *p* < 0.05 for the entire study. We separated patients into two groups: patients with (group R) and without (group NR) eye rubbing. 

## 3. Results

### 3.1. Characteristics of the Study Population

Between May and July 2021, 153 patients were included in the study: 69 keratoconus (45.1%) and 84 chronic ocular surface pathologies (54.9%). Of these patients, 125 (81.7%) reported rubbing their eyes. Figure 1 shows the study’s flow chart.

The average age was 39 ± 17 years and 61.4% of patients were female. Of the patients surveyed, 18 patients (11.8%) reported an addiction disorder and 38 patients (24.8%) reported a psychiatric disorder, mainly depression and anxiety. The characteristics of the study population are described in Table 1.

The KC group had a corrected distance visual acuity (CDVA) of 0.1 (±1.7 lines) LogMAR, a Kmax of 58.57D (±10.14) and a thinnest pachymetry of 431 μm (±49). On clinical examination, nine patients (13%) had corneal opacities and 32 patients (46.4%) had Vogt’s striae.

The OSD group included 77 Dry Eye Diseases (DED) (91.7%), 19 blepharitis (22.6%) and 14 chronic allergic conjunctivitis (16.7%). The CDVA was 0.02 (±0.6 line) LogMAR. Average BUT was 5.9 (±2.39) seconds and mean Oxf score was 0.81 (±1.04). Meibomian gland dysfunction (MGD) was found in 58.3% of patients.

### 3.2. Comparison of Keratoconus (KC) and Ocular Surface Disease (OSD) Groups

Several significant differences were found between the KC and OSD groups (Table 1). Mean age was 30 ± 9 years in the KC group and 46 ± 18 years in the OSD group (*p* < 0.001). Of the patients surveyed, 17.4% of KC and 38.1% of OSD reported a dermatological history (*p* = 0.005), 26.1% and 47.6%, respectively, experienced skin pruritus (*p* = 0.04), while 18.8% and 38.1%, respectively, reported skin scratching (*p* = 0.009).

Ocular pruritus was more frequent in KC, affecting 58% of patients, compared with 40.5% of OSD (*p* = 0.03). The modified SANDE score was higher for OSD, with a significant difference for frequency (*p* < 0.001) and intensity (*p* < 0.001) of symptoms experienced by patients. Finally, 62 (89.9%) KC and 63 (75%) OSD reported eye rubbing (*p* = 0.02). Ophthalmological examination found significantly more papillae in KC (*p* = 0.0007) and more MGD in OSD (*p* < 0.0001).

### 3.3. Comparison of Rubbing (R) and Non-Rubbing (NR) Groups

We compared patients with (group R) and without eye rubbing (NR group) (Table 2). The mean age was 46 ± 19 years in the NR group and 37 ± 16 years in the R group (*p* = 0.01). In the NR group, seven patients (25%) presented a keratoconus versus 62 (49.6%) in the R group (*p* = 0.02). In addition, 20 patients (71.4%) in the NR group had DED versus 61 (48.8%) in the R group (*p* = 0.03). The two groups showed no difference for anti-allergic treatment. Patients in the NR group were significantly more treated with anti-inflammatory eyedrops (*p* = 0.049). Dermatologically, 17.9% of NR patients experienced skin pruritus compared to 38.4% of R patients (*p* = 0.04), while 14.3% of NR patients reported scratching their skin compared to 32.8% of R patients (*p* = 0.052). Allergy was reported by seven NR patients (25%) and 65 R patients (52%) (*p* = 0.01). No significant difference was found for psychiatric and addictive history.

Among the patients interviewed, 75% of NR patients reported ocular symptoms compared to 92.8% of R patients (*p* = 0.006). Ocular pruritus was reported by 7.1% of NR patients and 57.6% of R patients (*p* < 0.0001). Slit lamp examination found papillae in six (21.4%) NR patients and 82 (65.6%) R patients (*p* < 0.0001). Mean BUT was 7.27 (±2.80) in NR and 6.14 (±2.68) in R (*p* = 0.047). Mean Oxford grade was 0.34 (±0.58) in NR and 0.59 (±0.95) in R (*p* = 0.011). Subpopulation analyses of KC with and without ocular rubbing showed no significant difference for clinical severity.

### 3.4. Eye Rubbing Analysis

Characteristics of eye rubbing were studied in 125 patients (Table 3). Nighttime rubbing was reported by 53.2% of KC patients versus 28.6% of OSD patients (*p* = 0.005), while 40.3% of KC patients and 71.4% of OSD patients reported rubbing their eyes regardless of the time of day (*p* < 0.002). Daily occurrence of eye rubbing was 32.3% for KC versus 55.6% for OSD (*p* = 0.009). Daily frequency was higher for OSD (*p* = 0.002).

Regarding Goodman criteria, the average score obtained for eye rubbers was 5.8 ± 3.1. Among these patients, 79 (63.2%) had a Goodman score of 5 and above. The CAGE questionnaire reported a mean score of 2.2 ± 1.2 for the rubbing population. The CAGE score greater than or equal to 2 was found in 93 patients (74.4%). KC and OSD subgroups did not differ significantly for Goodman criteria or CAGE score.

### 3.5. Comparison According to Goodman Criteria

We compared patients with Goodman scores of 5 and above (positive score) and those below 5 (negative score) (Table 4). Age was comparable between the two groups. No difference was found for ophthalmologic history. Existence of an addiction was reported by 16.5% of positive patients versus 4.3% of negative patients (*p* = 0.045). Psychiatric family history was reported by 17.7% of positive patients and 4.3% of negative patients (*p* = 0.03). There was no significant difference in personal psychiatric history between the two groups. Patients with a positive score had significantly more ocular symptoms (*p* = 0.008) and itching (*p* = 0.001) than those with a negative score. Similarly, the frequency and intensity of symptoms were higher in the positive population (*p* = 0.03 and *p* = 0.001, respectively). Finally, eye rubbing’s frequency and intensity were higher in the positive score population (*p* < 0.001). On clinical examination, 58 (70.7%) of the positive patients and 21 (48.8%) of the negative patients had papillae (*p* = 0.02). Mean BUT was significantly shorter in patients with a positive score (*p* = 0.03).

### 3.6. Intention to Stop after the Questionnaire

Of the 125 eye rubbers interrogated, 87 patients (69.6%) declared they would be willing to stop eye rubbing after completing the questionnaire. The positive responses rate was 85.5% for KC versus 54% for OSD (*p* = 0.0001). This rate was not different between patients with a positive and negative Goodman score.

## 4. Discussion

Our study showed that eye rubbing shares common characteristics with addictive behaviors. Our results showed that 63.3% of patients rubbing their eyes met at least five addiction diagnostic criteria according to Goodman. In addition, 74.4% of rubbers had a positive CAGE score predictive of addiction.

This is the first study to address eye rubbing as a cognitive–behavioral symptom. Therefore, results obtained cannot be compared with existing ophthalmological data. However, eye rubbing can be equated with skin scratching. Studies in dermatology found common neurobiological bases between the itch–scratch cycle and addictions [13,14].

In our study, the percentage of eye rubbing was high in the general population (81.1%), in KC (89.9%) and in OSD (75%). In comparison, a study found 29.5% eye rubbing in a population visiting an eye center in the United States [3]. This important difference with the literature can be explained by the selection of pathologies (KC and OSD) for which the proportion of eye rubbing is higher. Rates found for KC are in the high range of rubbing frequencies found in the literature [6]. This may be the result of a selection bias secondary to the recruitment of patients in a tertiary care center with more severe keratoconus. Indeed, the literature reports an association between severity of keratoconus and the extent of ocular rubbing [5,6]. Therefore, the advanced stage of patients evaluated in our study (Kmax: 58.6D; thinnest pachymetry: 431 μm) could explain the high eye-rubbing prevalence. As for OSD, a study reported 40% rubbing for DED, 51.8% for allergic conjunctivitis and 41.8% for blepharitis [3]. The higher rates found in our study could be once again secondary to a recruitment bias or to a collection facilitated by our questionnaire.

The severity of DED was higher in patients with eye rubbing (more frequent symptoms, shorter BUT and higher Oxford). Eye rubbing could act on ocular surface inflammation and cell apoptosis, which are essential elements of DED’s vicious cycle described by Baudouin et al. [8,18].

Ophthalmological symptoms reported were more intense and frequent for patients with higher Goodman score. Similarly, frequency and duration of eye rubbing were greater for these patients. This could be secondary to peripheral and central sensitization, which are phenomena well described in skin pruritus [19].

No significant difference was found between KC and OSD for diagnostic and predictive criteria of addiction. Eye rubbing was more frequent in KC but OSD had higher daily frequency. These results suggest that it is not the cognitive–behavioral characteristics of eye rubbing that are responsible for the occurrence of KC but rather its mechanical characteristics combined with corneal susceptibility to ectasia.

Conjunctival papillae seem to play an important role in eye rubbing. First, their presence was more frequent in KC as well as in patients rubbing their eyes. Second, in the eye rubbing population, papillae were significantly more frequent in patients with high addiction criteria scores. Papillae could not only be an essential element in the pathophysiology of eye rubbing but could also be an essential element in the eye rubbing cycle. Papillae could be part of the peripheral sensitization phenomenon in ocular pruritus. Further studies are needed to investigate this hypothesis.

Patients without eye rubbing were significantly more treated with anti-inflammatory eye drops (cyclosporine and corticosteroids) than eye rubbers, whereas there was no difference for antihistamine and mast cell stabilizer drops. These results support the hypothesis that chronic ocular pruritus, similar to skin pruritus, is primarily mediated by non-histaminergic pathways and is supported by local inflammatory mechanisms [20].

After answering the questionnaire, patients with KC were more likely to stop rubbing their eyes than those with OSD. These positive responses may be secondary to patients’ awareness of eye rubbing’s role in the pathophysiology of KC [21]. On the other hand, few data are available for patients and practitioners regarding eye rubbing in OSD. Appropriate information about eye rubbing and its consequences would raise awareness among patients and the general population. This could limit the occurrence and progression of keratoconus [9]. Similarly, educating patients with OSD could limit a potential aggravating factor of inflammation and DED [8,10,18].

In our study, patients with higher Goodman criteria scores reported more addictions and family psychiatric history. These results are consistent with data reporting the existence of psychiatric and addictive comorbidities concerning up to 50% of patients with addictive disorders [22,23].

The findings of this study have to be seen in light of some limitations. First, there was a selection bias related to recruiting patients from a tertiary care center, selecting more severe KC and OSD. In addition, the inclusion of KC or OSD in this study limits the extrapolation of the results to the general population. A controlled study with healthy eye rubbers and less severe patients would provide additional data. Second, there was an assessment bias inherent to answering a self-questionnaire with psychiatric questions. This could result in an underestimation of psychiatric and addictive disorders as a result of negative responses [24,25]. Therefore, it is possible that our study underestimates psychiatric and addictive comorbidities.

Finally, an important bias of this study is the lack of validated specific diagnostic criteria. Indeed, specific diagnostic criteria exist for substance addictions and some behavioral addictions, but no generic questionnaire is validated for behavioral addictions [26]. In this study, we used Goodman’s generic diagnostic criteria, which are not commonly used by psychiatrists, in order to adapt them to eye rubbing [16]. Therefore, we cannot conclude that eye rubbing is a behavioral addiction. However, we can affirm that eye rubbing and addiction share many cognitive and behavioral characteristics.

The parallel between eye rubbing and addiction allows us to consider new clinical and therapeutic approaches. Patient management should include information on ocular pruritus, eye rubbing and their potential consequences. Evaluation of ocular pruritus and rubbing in KC and OSD could allow a more appropriate management by involving patients and allowing better understanding of their symptoms, behavior and resulting disorders. As suggested by McMonnies, an informative document can be given to eye rubbers to limit the apparition and progression of corneal ectasia [27]. When the behavior is strong, behavioral substitution is possible by asking patients to rub their orbital rims instead of their eyes. Motivational interviewing and cognitive–behavioral therapy could be useful in severe cases as well as psychotropic treatments after discussion with a specialist (13). These therapeutic options target the cognitive and behavioral features of eye rubbing. However, it is important to combine this management with that of ocular pruritus. In addition to antihistamines and mast cell stabilizers, which are only partially effective, anti-inflammatory treatments such as local corticosteroids, cyclosporine and autologous serum can be used to modulate the non-histaminergic inflammatory component [3,20]. While waiting for the development of specific molecules targeting pruritus such as QX-314, the management of pruritus and eye rubbing must be personalized and adapted to each patient [11].

Dermatologists studied the central and peripheral mechanisms of skin pruritus and scratching, describing the “itch–scratch cycle” [12,13]. Following the example of these studies, many explorations are possible in ophthalmology in order to deepen the concept of the eye rubbing cycle. Functional imaging studies could highlight the activation of cortical reward pathways and allow the study of central sensitization phenomena [14]. Cognitive–behavioral studies based on hetero-evaluation would complete the information collected in this study (mirror phenomena, pleasure felt during rubbing, comparison of induced and spontaneous rubbing). In addition, confocal microscopy as well as cellular and molecular studies could help approaching the peripheral sensitization mechanism.

## 5. Conclusions

In conclusion, eye rubbing is a behavior with addictive-like cognitive–behavioral characteristics in patients with KC or OSD. Eye rubbing is part of a vicious cycle with peripheral and central sensitization. This cycle plays an essential role in the occurrence and progression of keratoconus. Moreover, it could constitute an important element of maintenance and aggravation of DED, explaining the high prevalence of rubbing found in our OSD population. 

## Figures and Tables

**Figure 1 jcm-12-01529-f001:**
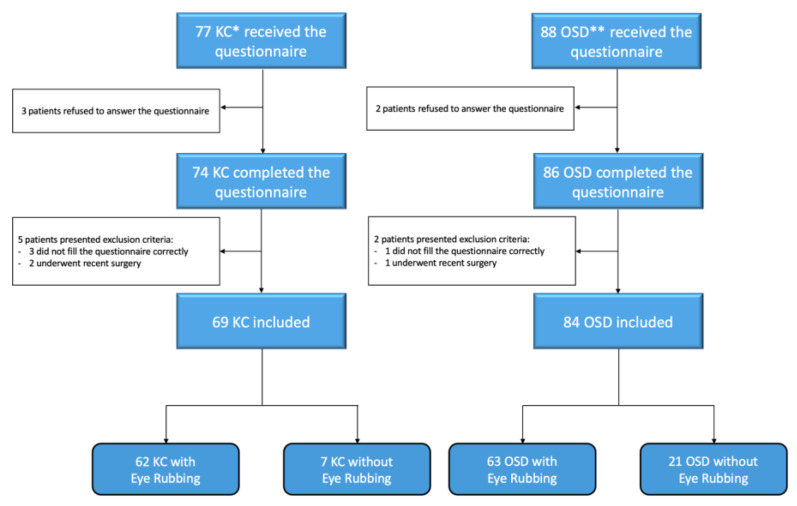
Flow chart of the study population. * KC: Keratoconus; ** OSD: Ocular Surface Disease.

**Table 1 jcm-12-01529-t001:** Characteristics of the study population: Keratoconus (KC) and Ocular Surface Disease (OSD) groups.

	Study Population (*n* = 153)	Keratoconus (*n* = 69)	Ocular Surface Disease (*n* = 84)	*p*
Age (years)	38.8 ± 17	30.1 ± 9	45.9 ± 18	≤0.001
Sex	94 F (61.4%)	37 F (53.6%)	57 F (67.9%)	0.07
Nighttime work	11 (7.2%)	8 (11.6%)	3 (3.6%)	0.06
Keratoconus	69 (45.1%)	69 (100%)	0 (0%)	≤0.0001
Dry eye disease	81 (52.9%)	4 (5.8%)	77 (91.7%)	≤0.0001
Allergic conjunctivitis	16 (10.5%)	2 (2.9%)	14 (16.7%)	0.006
Blepharitis	20 (13.1%)	1 (1.4%)	19 (22.6%)	<0.0001
**History of eye surgery**	34 (22.2%)	19 (27.5%)	15 (17.9%)	0.15
Cataract surgery	11 (7.2%)	0 (0%)	11 (13.1%)	0.002
Corneal graft	4 (2.6%)	3 (4.3%)	1 (1.2%)	0.22
Crosslinking	12 (7.8%)	12 (17.4%)	0 (0%)	<0.0001
Glasses	112 (73.2%)	50 (72.5%)	62 (73.8%)	0.86
Soft lenses	11 (7.2%)	1 (1.5%)	10 (11.9%)	0.01
Rigid lenses	19 (12.4%)	16 (23.2%)	3 (3.6%)	<0.001
**Ophthalmologic treatment**	96 (62.7%)	39 (56.5%)	57 (67.9%)	0.15
Artificial tears	86 (56.2%)	32 (46.4%)	54 (63.3%)	0.03
Antiallergic drops	35 (22.9%)	23 (33.3%)	12 (14.3%)	0.005
Anti-inflammatory drops	29 (19%)	4 (5.8%)	25 (29.8%)	<0.001
Dermatological history	44 (28.8%)	12 (17.4%)	32 (38.1%)	0.005
Allergy	72 (47.1%)	32 (44.4%)	40 (47.6%)	0.88
Skin pruritus	53 (34.6%)	18 (26.1%)	35 (41.7%)	0.04
Skin scratch	45 (29.4%)	13 (18.8%)	32 (38.1%)	0.009
Active smoking	27 (17.6%)	10 (14.5%)	17 (20.2%)	0.36
**Addictive history**	18 (11.8%)	5 (7.2%)	13 (15.5%)	0.12
Alcohol	5 (3.3%)	2 (2.9%)	3 (3.6%)	0.82
Tobacco	11 (7.2%)	1 (1.4%)	10 (11.9%)	0.01
Video games	3 (2%)	3 (4.3%)	0 (0%)	0.054
**Psychiatric history**	38 (24.8%)	13 (18.8%)	25 (29.8%)	0.12
Depression	21 (13.7%)	9 (13%)	12 (14.3%)	0.82
Anxiety	29 (19%)	9 (13%)	20 (23.8%)	0.09
Bipolar disorder	4 (2.6%)	1 (1.4%)	3 (3.6%)	0.41
Psychiatric follow-up	6 (3.9%)	2 (2.9%)	4 (4.8%)	0.56
Familial psychiatric history	21 (13.7%)	4 (5.8%)	17 (20.2%)	0.01
**Ocular symptoms**	137 (89.5%)	58 (84.1%)	79 (94%)	0.045
Ocular pruritus	74 (48.8%)	40 (58%)	34 (40.5%)	0.03
Ocular burning	42 (27.5%)	6 (8.7%)	36 (42.9%)	≤0.0001
Dryness	36 (23.5%)	8 (11.6%)	28 (33.3%)	0.002
Foreign body sensation	24 (15.7%)	10 (14.5%)	14 (16.7%)	0.71
Irritation	39 (25.5%)	16 (23.2%)	23 (27.4%)	0.56
Discomfort	22 (14.4%)	11 (15.9%)	11 (13.1%)	0.62
**SANDE score for symptoms**				
Frequency (/100)	55.07	44.37	63.48	≤0.001
Intensity (/100)	51.96	42.5	58.9	≤0.001
**Eye rubbing**	125 (81.7%)	62 (89.9%)	63 (75%)	0.02
CDVA * (LogMAR)	0,06	0.1	0.02	≤0.001
Papillae	88 (57.5%)	50 (72.5%)	38 (45.2%)	< 0.001
MGD **	54 (35.3%)	5 (7.2%)	49 (58.3%)	≤0.0001
Mean BUT ***	6.35	6.88	5.9	0.03
Mean Oxford grade	0.54	0.22	0.81	≤0.001

* CDVA: Corrected Distance Visual Acuity ** MGD: Meibomian Gland Dysfunction *** BUT: Break-Up Time.

**Table 2 jcm-12-01529-t002:** Characteristics of the study population: No Rubbing (NR) and Rubbing (R) groups.

	Study Population (*n* = 153)	No Rubbing (*n* = 28)	Rubbing (*n* = 125)	*p*
Age (years)	38.8	45.9	37.2	0.02
Sex	94 F (61.4%)	16 F (57.1%)	78 F (62.4%)	0.60
Nighttime work	11 (7.2%)	2 (7.1%)	9 (7.2%)	0.99
Keratoconus	69 (45.1%)	7 (25%)	62 (49.6%)	0.02
Dry eye disease	81 (52.9%)	20 (71.4%)	61 (48.8%)	0.03
Allergic conjunctivitis	16 (10.5%)	1 (3.6%)	15 (12%)	0.19
Blepharitis	20 (13.1%)	4 (14.3%)	16 (12.8%)	0.83
**History of eye surgery**	34 (22.2%)	7 (25%)	27 (21.6%)	0.69
Cataract surgery	11 (7.2%)	4 (14.3%)	7 (5.6%)	0.11
Corneal graft	4 (2.6%)	0 (0%)	4 (3.2%)	0.34
Crosslinking	12 (7.8%)	1 (3.6%)	11 (8.8%)	0.35
Glasses	112 (73.2%)	21 (75%)	91 (72.8%)	0.81
Soft lenses	11 (7.2%)	3 (10.7%)	8 (6.4%)	0.43
Rigid lenses	19 (12.4%)	1 (3.6%)	18 (14.4%)	0.12
**Ophthalmologic treatment**	96 (62.7%)	18 (64.3%)	78 (62.4%)	0.86
Artificial tears	86 (56.2%)	16 (57.1%)	70 (56%)	0.91
Antiallergic drops	35 (22.9%)	3 (10.7%)	32 (25.6%)	0.09
Anti-inflammatory drops	29 (19%)	9 (32.1%)	20 (16%)	0.049
Dermatological history	44 (28.8%)	10 (35.7%)	34 (27.2%)	0.37
Allergy	72 (47.1%)	7 (25%)	65 (52%)	0.009
Skin pruritus	53 (34.6%)	5 (17.9%)	48 (38.4%)	0.04
Skin scratch	45 (29.4%)	4 (14.3%)	41 (32.8%)	0.053
Active smoking	27 (17.6%)	5 (17.9%)	22 (17.6%)	0.97
**Addictive history**	18 (11.8%)	3 (10.7%)	15 (12%)	0.85
**Psychiatric history**	38 (24.8%)	7 (25%)	31 (24.8%)	0.98
Depression	21 (13.7%)	5 (17.9%)	16 (12.8%)	0.48
Anxiety	29 (19%)	5 (17.9%)	24 (19.2%)	0.87
Bipolar disorder	4 (2.6%)	2 (7.1%)	2 (1.6%)	0.097
Psychiatric follow-up	6 (3.9%)	2 (7.1%)	4 (3.2%)	0.33
Familial psychiatric history	21 (13.7%)	5 (17.9%)	16 (12.8%)	0.48
**Ocular symptoms**	137 (89.5%)	21 (75%)	116 (92.8%)	0.0056
Ocular pruritus	74 (48.8%)	2 (7.1%)	72 (57.6%)	<0.0001
Ocular burning	42 (27.5%)	11 (39.3%)	31 (24,8%)	0.12
Dryness	36 (23.5%)	8 (28.6%)	28 (22.4%)	0.49
Foreign body sensation	24 (15.7%)	3 (10.7%)	21 (16.8%)	0.43
Irritation	39 (25.5%)	7 (25%)	32 (25.6%)	0.95
Discomfort	22 (14.4%)	2 (7.1%)	20 (16%)	0.23
**SANDE score for symptoms**				
Frequency (/100)	55.07	59.24	54.78	0.50
Intensity (/100)	51.96	59.95	51.05	0.36
CDVA * (LogMAR)	0.06	0.04	0.06	0.34
Papillae	88 (57.5%)	6 (21.4%)	82 (65.6%)	<0.0001
MGD **	54 (35.3%)	17 (60.7%)	37 (29.6%)	0.0019
Mean BUT ***	6.35	7.27	6.14	0.047
Mean Oxford grade	0.54	0.34	0.59	0.01

* CDVA: Corrected Distance Visual Acuity ** MGD: Meibomian Gland Dysfunction *** BUT: Break-Up Time.

**Table 3 jcm-12-01529-t003:** Cognitive–behavioral characteristics of eye rubbing.

	Rubbing (*n* = 125)	Keratoconus (*n* = 62)	Ocular Surface Disease (*n* = 63)	*p*
**Eye rubbing characteristics**				
Occasional	63 (50.4%)	37 (59.7%)	26 (41.3%)	0.04
Daily	55 (44%)	20 (32.3%)	35 (55.6%)	0.009
Daily frequency (/100)	42.5	35.66	49.22	0.002
Duration (/100)	37.36	35.58	39.11	0.46
More frequent during morning	11 (8.8%)	8 (12.9%)	3 (4.8%)	0.11
More frequent at night	51 (40.8%)	33 (53.2%)	51 (28.6%)	0.005
No preference for period of day	70 (56%)	25 (40.3%)	45 (71.4%)	≤0.001
**Goodman criteria**				
Goodman criteria score (/16)	5.82	5.58	6.06	0.38
Goodman ≥ 5	79 (63.2%)	39 (62.9%)	40 (63.5%)	0.95
**CAGE score**				
Mean CAGE	2.23	2.34	2.13	0.33
CAGE ≥ à 2	93 (74.4%)	49 (79%)	44 (69.8%)	0.24
**Intention to stop**	87 (69.6%)	53 (60.9%)	34 (39.1%)	≤0.0001

**Table 4 jcm-12-01529-t004:** Comparison of patients according to Goodman’s criteria score.

	Rubbing (*n* = 125)	Goodman Score < 5 (*n* = 46)	Goodman Score ≥ 5 (*n* = 79)	*p*
Age (years)	37.2	35.9	38	0.48
Sex	78 F (62.4%)	24 F (52.2%)	54 F (68.4%)	0.07
Keratoconus	62 (49.6%)	23 (50%)	39 (49.4%)	0.95
Dry eye disease	61 (48.8%)	22 (47.8%)	39 (49.4%)	0.87
Allergic conjunctivitis	15 (12%)	6 (40%)	9 (11.4%)	0.07
Blepharitis	16 (12.8%)	7 (15.2%)	9 (11.4%)	0.54
**History of eye surgery**	27 (21.6%)	9 (19.6%)	18 (22.8%)	0.67
Glasses	91 (72.8%)	32 (69.6%)	59 (74.7%)	0.54
Soft lenses	8 (6.4%)	5 (11.1%)	3 (3.8%)	0.11
Rigid lenses	18 (14.4%)	8 (17.4%)	10 (12.7%)	0.47
**Ophthalmologic treatment**	78 (62.4%)	30 (65.2%)	48 (60.8%)	0.62
Artificial tears	70 (56%)	28 (60.9%)	42 (53.2%)	0.40
Antiallergic drops	32 (25.6%)	10 (21.7%)	22 (27.8%)	0.45
Anti-inflammatory drops	20 (16%)	9 (19.6%)	11 (13.9%)	0.41
Dermatological history	34 (27.2%)	8 (17.4%)	26 (32.9%)	0.06
Allergy	65 (52%)	23 (50%)	42 (53.2%)	0.73
Skin pruritus	48 (38.4%)	13 (28.3%)	35 (44.3%)	0.08
Skin scratch	41 (32.8%)	12 (26.1%)	29 (36.7%)	0.22
Active smoking	22 (17.6%)	5 (10.9%)	17 (21.5%)	0.13
**Addictive history**	15 (12%)	2 (4.3%)	13 (16.5%)	0.045
**Psychiatric history**	31 (24.8%)	8 (17.4%)	23 (29.1%)	0.15
Depression	16 (12.8%)	3 (6.5%)	13 (16.5%)	0.11
Anxiety	24 (19.2%)	7 (15.2%)	17 (21.5%)	0.39
Bipolar disorder	2 (1.6%)	0 (0%)	2 (2.5%)	0.28
Psychiatric follow-up	4 (3.2%)	0 (0%)	4 (5.1%)	0.12
Familial psychiatric history	16 (12.8%)	2 (4.3%)	14 (17.7%)	0.03
**Ocular symptoms**	116 (92.8%)	39 (84.8%)	77 (97.5%)	0.008
Ocular pruritus	72 (57.6%)	18 (39.1%)	54 (68.4)	0.002
Ocular burning	31 (24.8%)	10 (21.7%)	21 (26.6%)	0.55
Dryness	28 (22.4%)	10 (21.7%)	18 (22.8%)	0.89
Foreign body sensation	21 (16.8%)	11 (23.9%)	10 (12.7%)	0.11
Irritation	32 (25.6%)	8 (17.4%)	24 (30.4%)	0.11
Discomfort	20 (16%)	8 (17.4%)	12 (15.2%)	0.75
**SANDE score for symptoms**				
Frequency (/100)	54.78	46.74	58.91	0.03
Intensity (/100)	51.05	39.21	57.05	0.001
CDVA * (LogMAR)	0.06	0.06	0.06	0.89
Papillae	79 (63.2%)	21 (48.8%)	58 (70.7%)	0.02
MGD **	37 (29.6%)	16 (34.8%)	21 (26.6%)	0.33
Mean BUT ***	6.14	6.8	5.75	0.03
Mean Oxford grade	0.59	0.48	0.65	0.33
**Eye rubbing characteristics**				
Frequency per day (/100)	42.5	29.7	49.9	≤0.001
Duration (/100)	37.36	24.37	44.92	≤0.001
More frequent during morning	11 (8.8%)	4 (8.7%)	7 (8.9%)	0.98
More frequent at night	51 (40.8%)	19 (41.3%)	32 (40.5%)	0.93
No preference for period of day	70 (56%)	25 (54.3%)	45 (57%)	0.78
**Intention to stop**	87 (69.6%)	32 (69.6%)	55 (69.6%)	0.99

* CDVA: Corrected Distance Visual Acuity ** MGD: Meibomian Gland Dysfunction *** BUT: Break-Up Time.

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
