# Peer review of "EYERUBBICS: The Eye Rubbing Cycle Study"

_jcm, 2023, doi:10.3390/jcm12041529_

Round 1

Reviewer 1 Report

very well written manuscript.

Minor revisions

Student t-test, ANOVA test, and logistic 103

cyclosporine and autologous serum can be 293

-------------

The  Discussion

Our study showed that eye rubbing shares common characteristics with addictive behaviours. 206  

-------------

Line ---> 212. However, eye rubbing can be equated with skin scratching. Studies in dermatology found common neurobiological bases between the itch-scratch cycle and addictions [13,14].

In this paragraph you mention dermatology but you only talk about it at the end. An option would be to merge both paragraphs or add this phrase later in the last paragraph

 MERGE WITH Dermatologists studied the central and peripheral mechanisms of skin pruritus and scratching, describing the "itch-scratch cycle" [12,13]. Following the example of these study (Line 298)

-------------

Line 250 ---> Conjunctival papillae seem to play an important role in eye rubbing. First, their presence was more frequent in KC as well as in patients rubbing their eyes. Second, in the eye rubbing population, papillae were significantly more frequent in patients with high addiction criteria scores. Papillae could not only be an essential element in the pathophysiology of eye rubbing, but could also be an essential element in the eye rubbing cycle. Papillae could be part of the peripheral sensitization phenomenon in ocular pruritus. Further studies are needed to investigate this hypothesis. 

It would be great to have some references to this paragraph

-------------

In our study, patients with higher Goodman criteria scores reported more addictions and family psychiatric history. These results are consistent with data reporting the existence of psychiatric and addictive comorbidities concerning up to 50% of patients with addictive disorders [19,20]. 234

This paragraph is in the middle of different paragraphs that focus only in the eye

One option would be to put this paragraph following the first paragraph of the discussion

In our study, patients with higher Goodman criteria scores reported more addictions and family psychiatric history. These results are consistent with data reporting the existence of psychiatric and addictive Comorbidities concerning up to 50% of patients with addictive disorders [19,20]. 234

and then merge the other ones that talk only about eyes

-------------

Another way to structure the discussion would be talking about eyes and then about dermatology and psychiatry

you could add  this paragraph in the discussion 

Our study showed that eye rubbing shares common characteristics with addictions. Our results showed that 63.3% of patients rubbing their eyes met at least 5 addiction diagnostic criteria according to Goodman. In addition, 74.4% of rubbers had a positive CAGE score predictive of addiction. This is the first study to address eye rubbing as a cognitive-behavioral symptom. 210
Therefore, results obtained cannot be compared with existing ophthalmological data. 
However, eye rubbing can be equated with skin scratching. Studies in dermatology found common neurobiological bases between the itch-scratch cycle and addictions [13,14].

-------------

Patients with Down syndrome have more OCD and KTC, blepharitis and more, for example. Some of the patients in your study had Down? Do you think it is relevant to the discussion?

Author Response

Dear reviewer,

Thank you for your constructive review and comments. 

I took into consideration each of your remarks and modified my manuscript accordingly. As for the discussion, I took your comments into consideration to modifiy the text. 

Concerning Down syndrome, no patients included in this study presented this syndrome. 

Thank you for your time. 

Sincerely,

Dr Alexandre HAGE

Reviewer 2 Report

Line 18.

It reads: “Methods: A single-center prospective study was conducted in a tertiary ophthalmology center between May and July 2021.”

Comment:

Consider modifying to: “Methods: A prospective study was conducted in a tertiary ophthalmology center between May and July 2021.”

Line 44.

It reads: “The prevalence of eye rubbing in keratoconus varies between 66 and 91% [6].”

Comment:

I think it is warranted to mention here that eye rubbing has been implicated as a potential causative factor in keratoconus.

Then consider modifying to: "The prevalence of eye rubbing in keratoconus varies between 66 and 91% and it has been suggested that it could in fact be a causal factor of the disease [6].”

Additional references:

Galvis V, Tello A, Carreño NI, Berrospi RD, Niño CA. Risk Factors for Keratoconus: Atopy and Eye Rubbing. Cornea. 2017;36(1):e1. doi:10.1097/ICO.0000000000001052

Gatinel D, Tello A, Niño C, Translateur A, Lugo IK, Berrospi RD. Comment on Personalized Model to Predict Keratoconus Progression From Demographic, Topographic, and Genetic Data [published online ahead of print, 2022 Nov 12]. Am J Ophthalmol. 2022;S0002-9394(22)00427-5. doi:10.1016/j.ajo.2022.10.025

Bitton K, Dubois M, Moran S, Gatinel D. Discordant Keratoconus in Monozygotic Twins. Case Rep Ophthalmol. 2022;13(1):313-317. Published 2022 Apr 25. doi:10.1159/000524116

Mazharian A, Panthier C, Courtin R, et al. Incorrect sleeping position and eye rubbing in patients with unilateral or highly asymmetric keratoconus: a case-control study. Graefes Arch Clin Exp Ophthalmol. 2020;258(11):2431-2439. doi:10.1007/s00417-020-04771-z

Line 66:

It reads: “The non-inclusion criteria were as follows…”

Comment:

Consider modifying to: “The exclusion criteria were as follows…”

Line 97:

It reads: “best corrected visual acuity  (BCVA),”

Comment:

Consider modifying to: “ corrected  distance visual acuity  (CDVA),”. This adjustment must be made throughout the entire manuscript.

Line 100:

It reads: “the maximum keratometry (Kmax) and the thinnest pachymetry.”

Comment:

It is necessary to indicate what device was used to perform the corneal measurements.

Table 1.

It reads:

Age (years)

38.8

30.1

45.9

Comment

It would be preferable to include also the standard deviation of each group´s age.

Author Response

Dear reviewer,

Thank you for your time to review my paper and for ther constructive comments made. I've taken all your remarks into considerations when changing my manuscript. 

Sincerely,

Dr Alexandre HAGE